# Psoralen Alleviates Renal Fibrosis by Attenuating Inflammasome-Dependent NLRP3 Activation and Epithelial–Mesenchymal Transition in a Mouse Unilateral Ureteral Obstruction Model

**DOI:** 10.3390/ijms241713171

**Published:** 2023-08-24

**Authors:** Tae Won Lee, Eunjin Bae, Jin Hyun Kim, Myeong Hee Jung, Dong Jun Park

**Affiliations:** 1Department of Internal Medicine, Gyeongsang National University Changwon Hospital, Changwon 51353, Republic of Korea; milkey@hanmail.net (T.W.L.); delight7607@naver.com (E.B.); 2Department of Internal Medicine, Gyeongsang National University College of Medicine, Jinju 52828, Republic of Korea; 3Institute of Medical Science, Gyeongsang National University, Jinju 52828, Republic of Korea; ajini7044@hanmail.net (J.H.K.); yallang7@hanmail.net (M.H.J.); 4Biomedical Research Institute, Gyeongsang National University Hospital, Jinju 52828, Republic of Korea

**Keywords:** renal fibrosis, psoralen, EMT, inflammation

## Abstract

The role of psoralen (PS), a major active component extracted from *Psoralea corylifolia* L. seed, in renal fibrosis is still unclear. Thus, the objective of this study was to evaluate the effects of PS on the development and progression of renal fibrosis induced by unilateral ureteral obstruction (UUO) in a mouse model. Mice were divided into four groups: PS (20 mg/kg, i.g., *n* = 5), PS + sham (*n* = 5), UUO (*n* = 10), and PS + UUO (*n* = 10). PS was intragastrically administered 24 h before UUO and continued afterwards for 7 days. All mice were killed 7 days post UUO. Severe tubular atrophy, tubular injury, and tubulointerstitial fibrosis (TIF) were significantly developed in UUO mice. A higher expression of transforming growth factor-β1 (TGF-β1) was accompanied by elevated levels of α-smooth muscle actin (α-SMA) and phosphorylated Smad2/3 (pSmad2/3) at 7 days post UUO. However, PS treatment reduced tubular injury, interstitial fibrosis, and the expression levels of TGF-β1, α-SMA, and pSmad2/3. Furthermore, the levels of macrophages (represented by F4/80 positive cells) and the inflammasome, reflected by inflammasome markers such as nucleotide-binding and oligomerization domain-like receptors protein 3 (NLRP3) and cleaved caspase1 (cCASP-1), were significantly decreased by PS treatment. These results suggest that PS merits further exploration as a therapeutic agent in the management of chronic kidney disease (CKD).

## 1. Introduction

Chronic kidney disease (CKD) is recognized as a global public health problem with significant morbidity and mortality in the general population [1,2,3]. Although a comprehensive understanding of the mechanisms underlying the progression of CKD remains to be elucidated, several studies have reported that hypertension (HT), proteinuria, obesity, diabetes, hypoxia–ischemia from vascular disease, and dyslipidemia can result in monocyte–macrophage recruitment to the kidney and chronic tubulointerstitial inflammation, which can lead to the progressive fibrosis. This is generally known as an excessive deposition of extracellular matrix components, which is considered to be the devastating final common pathway in most progressive renal diseases [4,5,6,7,8]. In CKD, targeting fibrotic progression may reduce renal injury and fibrosis reversal. Slowing down its progression might be a good therapeutic option for the treatment of CKD [9,10].

Renal inflammatory responses commonly occur in response to pathogens and sterile stimuli such as necrosis induced by ischemia, high glucose levels, and lipids [11,12]. The involvement of mononuclear inflammatory cells in damaged renal parenchyma is a common finding in failing kidneys. It correlates inversely with kidney function [13,14,15]. Sterile damage-associated molecular patterns (DAMPS) are cellular contents released after loss of plasma membrane integrity. They could trigger inflammation and induce the production of inflammatory cytokines and chemokines [16,17]. These DAMPS can activate the innate immune system in response to cellular injury and induce a proinflammatory response to help repair damaged tissues. The nucleotide-binding and oligomerization domain (NOD)-like receptors (NLRs) protein 3 (NLRP3) inflammasome is a multi-protein complex and sensor in innate immune cells activated by DAMPs. It promotes the secretion of pro-inflammatory cytokines, such as IL-1-β and IL-18. Several studies have demonstrated that the activation of the NLRP3 inflammasome is associated with pathogenesis inflammation and fibrosis in several CKD models [18,19,20].

Psoralen (PS) is a principal bioactive component of *Cullen corylifolium* (L.) Medik. It is also present in many vegetables and fruits, such as celery (*Apium graveolens* L.) and the common fig (*Ficus carica* L.). PS is a tricyclic coumarin-like aromatic compound with a molecular structure of 7H-Furo[3,2-g]benzopyran-7-one (molecular weight: 186.16; molecular formula: C_3_H_6_O_3_). PS has been used in herbal and traditional medicine for various diseases, including diabetes, cancer, inflammatory disease, neurodegenerative disease, and osteoporosis [21,22]. PS also has protective effects on oxidative-stress-induced pancreatic beta cell apoptosis [23] and hepatic damage [24]. The protective effect of PS in bleomycin-induced pulmonary fibrosis in mice was recently reported [25]. In this study, we demonstrate that PS could alleviate renal interstitial fibrosis by attenuating inflammasome-dependent NLRP3 activation and epithelial–mesenchymal transition (EMT) in a mouse unilateral ureteral obstruction (UUO) model of CKD. We employed UUO which has become a representative model of tubulointerstitial renal fibrosis in a short time span, as it shows the fundamental pathogenetic mechanisms that express all forms of CKD. The results of this study provide insights into the processes driving renal inflammation and CKD progression and further identify PS as having therapeutic potential when used to combat CKD progression and encourage reversal.

## 2. Results

### 2.1. PS Improves Tissue Injury and Fibrosis in the UUO Mouse Model

H&E staining was performed to examine tissue injury. A normal renal cortex was found in the kidneys of sham-operated (sham) mice and mice treated with PS only. In the UUO group, there was severe tubular atrophy, tubular expansion, epithelial cell swelling, and inflammatory cell infiltration in the interstitial space (Figure 1).

Tissue fibrosis was investigated using MT staining. Both the sham and PS groups showed a normal renal cortex and interstitium without fibrosis (blue color). However, obvious interstitial fibrosis was observed in the UUO group, with collagen deposited in the interstitial space (Figure 2). UUO kidneys showed severe interstitial fibrosis and collagen deposition, whereas kidneys in the UUO + PS group only showed mild fibrosis at 7 days after UUO surgery (Figure 2).

### 2.2. PS Reduces Tubular Epithelial Cell Apoptosis and Macrophage Infiltration Caused by UUO

In UUO mice, tubular injury and cell death were induced by marked hemodynamic and metabolic changes. As a result, apoptosis with interstitial macrophage infiltration was observed in tubular epithelial cells. Apoptosis was confirmed through TUNEL staining. It was significantly increased in the UUO group compared to that in the sham group or the PS group. Apoptosis of tubular epithelial cells was reduced when the UUO group was treated with PS (Figure 3).

The interstitial inflammatory response was determined in this study. It was evaluated based on macrophage infiltration in the interstitium. The macrophage population was determined using F4/80 staining. It showed a significant increase in the UUO group compared with that in the sham control group and the PS treatment group. However, PS treatment significantly alleviated macrophage infiltration in the UUO group (Figure 4).

### 2.3. PS Attenuates the NLRP3 Inflammasome Induced by UUO

To evaluate the effect of PS on NLRP3 inflammasome activation, levels of NLRP3 inflammasome, caspase-1, mature IL-1β, and IL-18 were investigated. When evaluated through Western blot analysis, NLRP3 inflammasome and cleaved caspase-1 levels were increased after UUO surgery. Such increases were significantly attenuated in PS-treated UUO mice (Figure 5A,B). To confirm the effects of PS on inflammasome inactivation in UUO mice, mRNA expression levels of IL-1β and IL-18 (pro-inflammatory cytokines related to the NLRP3 inflammasome) were investigated. IL-1β and IL-18 mRNA expression levels were increased in UUO-only mice. However, they were decreased by PS treatment (Figure 5C). To verify the capacity to prove that macrophage infiltration occurs dependent on NLRP3 activation, we performed double immunohistochemical staining for macrophage (F4/80) and NLRP3. As shown in Figure 5D, mostly NLRP3-positive signals were detected in injured tubular epithelial cells and minorly macrophages and the double-positive signals for macrophages and NLRP3 were decreased by PS treatment. PS, itself, decreased the macrophage infiltration and also NLRP3 expression on the macrophages. Therefore, this result shows that PS reduces NLRP3 expression in macrophages as well as NLRP3 expression in tubular epithelial cells in the UUO model.

### 2.4. PS Inhibits Activation of TGF-β1/Smad 2/3 Signaling and EMT Pathway

It is well known that tubulointerstitial fibrosis (TIF) proceeds through the activation of TGF-β1 and the activation of Smad 2/3, a downstream signaling system [26,27,28]. Using immunoblot, we evaluated the levels of TGF-β1, phosphorylated Smad 2/3, and profibrotic proteins, such as α-SMA. As expected, TGF-β1, Smad 2/3 signaling, and α-SMA were significantly increased in the UUO model. However, PS treatment markedly decreased the expression levels of TGF-β1 and Smad 2/3 signaling in UUO mice (Figure 6). These results suggest that the activation of the TGF-β1 and Smad 2/3 pathways and α-SMA, a profibrotic protein, can be downregulated by PS treatment. Immunohistochemical staining of α-SMA, a myofibroblast marker, was performed to confirm fibrosis. α-SMA-positive signals were found in the tubulointerstitial areas of kidneys that underwent UUO. The number of positive signals was significantly reduced in PS-treated UUO mice (Figure 6). EMT is an important factor in the development and progression of TIF. Expression levels of TGF-β1 and Snail mRNA (i.e., EMT-specific biomarkers) were significantly reduced in PS-treated UUO mice (Figure 6).

## 3. Discussion

The present study demonstrated that PS could ameliorate renal interstitial fibrosis by attenuating inflammasome-dependent NLRP3 activation and EMT. Our results show that NLRP3 activation was associated with renal fibrosis. Our results also suggest that it could mediate EMT induced by TGF-β. The effects of NLRP3 on TGF-β signaling occurred through caspase-1, IL-1-β, and IL-18 activation. PS reduced these pathologic findings by interrupting these pathways. Thus, this merits the further evaluation of PS as a therapeutic option for CKD.

Persistent inflammation significantly contributes to renal fibrosis. Many researchers have reported the important role of NLRP3 inflammasome in diabetic kidney disease, hypertensive renal disease, ischemic renal disease, and UUO [29,30,31,32]. UUO is a model of CKD caused by progressive tubulointerstitial fibrosis. To investigate the role of NLRP3 in renal fibrosis, NLRP3 knockout (KO) mice have been used [9,29,33]. NLRP3 KO mice showed less renal inflammation and fibrosis at 14 days after UUO in comparison with normal mice without NLRP3 KO. These findings were associated with the activation of the inflammasome-dependent pathway of NLRP3 [29]. This study also showed that the NLRP3 mRNA level was increased in human fibrotic renal disease and positively correlated with serum creatinine levels. Another study demonstrated that the neutralization of IL-18, a downstream cytokine of the inflammasome-dependent pathway of NLRP3, can prevent renal fibrosis after UUO in mice [34]. In the present study, we found that the activation of the NLRP3 inflammasome and its downstream cytokines and renal fibrosis were increased at 7 days after UUO. However, they were downregulated after treatment with PS. These findings are consistent with those of previous studies, showing that renal fibrosis was also associated with inflammasome-dependent NLRP3 activation [9,29,33,34]. This present study is unique in that it suggests the potential of applying PS as a treatment for renal fibrosis in an in vivo model.

In addition to the inflammasome-dependent pathway, NLRP3 possesses biologic functions in various kidney diseases without forming an inflammasome, which is called the inflammasome-independent pathway of NLRP3 [33,35,36], the exact mechanisms of which have not yet been revealed. Blockade of IL-1β, the end product of an activated NLRP3 inflammasome, could not mitigate kidney diseases. Therefore, interest in the inflammasome-independent pathway of NLRP3 has increased [37,38,39]. NLRP3 is associated with renal fibrosis after UUO, whereas IL-1β, IL-18, and caspase-1 are not necessary for these pathways [38]. NLRP3 can also regulate mitochondrial reactive oxygen species (ROS) via TGF-β signaling and fibrosis, independent from the inflammasome [39]. NLRP3 inhibition, but not IL-1 receptor antagonists, can affect the shift of renal resident mononuclear cells (RMNCs) such as macrophages and dendritic cells to an anti-inflammatory phenotype. Primary murine embryonic fibroblasts and primary renal fibroblasts do not release IL-1β after LPS and ATP stimulation, although these cells also contain NLRP3 [35]. The process by which NLRP3 contributes to renal injury and progression is presumed to involve a combination of inflammasome-dependent and independent pathways [36,37]. The present study was focused on the beneficial effect of PS involving the inflammasome-dependent pathway of NLRP3.

The expression of NLRP3 was found in various components of renal cells, including renal RMNCs, renal tubular epithelial cells (TECs), renal fibroblasts, podocytes, glomerular endothelial cells, and mesangial cells [31,36]. However, in terms of the expression of active caspase-1 and the secretion of IL-1β, renal TECs failed to induce IL-1β or IL-18 release [38,40]. Considering these data, we could conjecture that the NLRP3 in renal tubular cells could directly act as an inflammasome-independent protein. On the other hand, injury of renal TECs by a variety of insults can result in the release of endogenous cellular components and lead to the activation of the NLRP3 inflammasome [41]. Furthermore, renal fibroblasts also express NLRP3 without secreting IL-1β upon LPS and ATP treatment [35]. NLRP3 in renal fibroblasts might induce tubulointerstitial fibrosis via TGF-β signaling in an inflammasome-independent pathway [35]. On the contrary, the NLRP3 inflammasome in renal RMNCs or infiltrating leukocytes also regulates IL-1β and IL-18, which contribute to tubular injury and fibrosis [34,42]. Our results also showed macrophage activation, NLRP3 activation, damaged renal tubular cells, and renal fibrosis due to UUO. We believe that the coordination of these immune and non-immune cells might play pivotal roles in renal fibrosis through both inflammasome-dependent and -independent pathways.

EMT is one of the mechanisms that can induce renal interstitial fibrosis [43]. In the process of fibrosis, tubular epithelial cells have the capacity to acquire an EMT in the injured kidney through the most important factor, TGF-β [44]. Recent studies have revealed that TGF-β activation can induce NLRP3 abundance in a Smad3-dependent manner and that NLRP3 can also promote the progression of renal tubular EMT by enhancing TGF-β signaling and activating receptor-regulated Smad (R-Smad) [45,46]. Wang et al. also demonstrated that TGF-β can act as a priming signal and lead to NLRP3 protein induction. The activated signal can lead to caspase-1 activation and the following EMT. They also revealed that pterostilbene could prevent renal fibrosis by attenuating NLRP3 inflammasome activation and EMT [47]. Our data were consistent with previous studies, showing that TGF-β could promote the activation of NLRP3 inflammasome and that pro-caspase-1 was converted to its active form, cleave-caspase-1, in the obstructed kidney, as demonstrated by Western blot analysis.

PS has diverse pharmacological benefits for the treatment of a variety of diseases, such as diabetes, cancer, inflammatory disease, neurodegenerative disease, and osteoporosis [21,22]. No reports have described the mechanism for a specific PS-mediated NLRP3 activation. One study showed that the activation of NLRP3 inflammasome played an important role in DSS-induced colonic inflammation damage [48]. PAP-1, also named 5-(4-phenoxybutoxy) PS, might mitigate the severity of colitis in mice via inhibition of the NLRP3 inflammasome pathway. These data are consistent with our results. Our study for the first time adopted the intragastric injection of PS in UUO-induced renal fibrosis and found that PS could alleviate renal inflammation and fibrosis in mice by inhibiting the NLRP3 pathway. No relevant data for PS-mediated NLRP3 inactivation are available yet in CKD.

In the present study, we first revealed that PS could be a potential agent for managing renal fibrosis and slowing the progression of CKD, although detailed mechanisms were not revealed. Plausible mechanisms are shown below. First, PS itself might directly and/or indirectly protect tubular cells and lead to the release of fewer endogenous molecules for activating the NLRP3 inflammasome. Second, PS might reduce the infiltration of inflammatory cells or the activation of renal RMNCs expressing F4/80 into the interstitium, thus mitigating the expression of TGF-β. Third, PS might mitigate inflammasome-dependent NLRP3 by interrupting steps such as the cleavage of pro-caspase-1 into caspase-1 and expressions of IL-1β and IL-18 mRNA, NLRP3-dependent cytokines. Fourth, PS might reduce renal fibrosis in inflammasome- and cytokine-independent fashions, although this mechanism was not confirmed in our study.

## 4. Materials and Methods

### 4.1. Animals, Surgery, and Tissue Preparation

Male C57BL/6 mice (10 weeks of age) were maintained in a temperature- and humidity-controlled facility with a 12 h/12 h light/dark cycle. Standard mice chow and water were provided ad libitum. Mice were assigned to four groups: a sham control group (*n* = 5 mice), a PS group (20 mg/kg, i.g., *n* = 5 mice), a unilateral ureteral obstruction (UUO) group (*n* = 10 mice), and a UUO + PS group (*n* = 10 mice). PS was intragastrically administered 24 h before UUO and continued afterwards for 7 days. Two individual experiments were conducted for the current study.

Mice were anesthetized with an intraperitoneal injection of Avertin (2,2,2-tribromoethanol, Sigma, St. Louis, MO, USA) to build a renal fibrosis model. After a left flank incision in the UUO group, complete ligation was performed using double silk sutures at the ureteropelvic junction of the left ureter. The contralateral kidney was also exposed. However, ureteral ligation was not conducted. In the sham control group, a left flank incision was made and both the kidney and ureter were exposed. However, the ureter was not ligated. These animals were sacrificed on day 7 post sham/UUO surgery. Blood and kidney tissues were harvested.

### 4.2. Renal Pathology

Kidneys were routinely fixed in 4% phosphate-buffered paraformaldehyde and embedded in paraffin. Tissue sections of 5 μm in thickness were obtained. Paraffin wax was removed with xylene and sections were rehydrated with ethanol. After washing, sections were stained with hematoxylin and eosin (H&E) and Masson trichrome (MT). H&E staining was performed for histopathological analysis. MT staining was used to assess tissue fibrotic changes. Semi-quantitative scoring was performed for H&E staining to examine the degree of interstitial injury, which was assigned points (0 to 3) based on the extent of interstitial fibrosis, tubular atrophy (defined as luminal dilation and flattened tubular epithelial cells), and interstitial inflammatory cell infiltration. Tissue injury (interstitial fibrosis, tubular atrophy, and interstitial inflammatory cell infiltration) was scored by grading the affected percentage under a high-powered field (×400) with minor modifications from a previous study [49]: score of 0, 0%; score of 1, <30%; score of 2, 31 to 60%; and score of 3, 61 to 100%. All scores were summed and are presented as average values in the presented graphs. Signals were analyzed using NIS-Elements BR 3.2 (Nikon, Tokyo, Japan).

### 4.3. Terminal Deoxynucleotidyl Transferase (TdT) dUTP Nick-End Labeling (TUNEL) Assay

TUNEL assays were conducted to evaluate apoptotic cells in kidney tissues using a commercial TUNEL assay kit (Roche Applied Sciences, Indianapolis, IN, USA) following the manufacturer’s instructions. For H&E staining, paraffin wax was removed using xylene. Tissues were then rehydrated and incubated with Proteinase K at 25 °C for 15–30 min. After washing twice with PBS, each section was added to a wet box at 37 °C for 1 h with 100 μL of TdT reaction solution. Each section was then washed three times with PBS and incubated for 1 h in streptavidin fluorescein–dUTP reaction solution at 37 °C. Semiquantitative analysis was performed by counting the number of TUNEL-positive cells per field in the renal tissue at ×400 magnification. At least 10 areas were randomly selected in the cortex per slide. The mean number of brown-colored cells in the selected field was expressed as the density of TUNEL-positive cells. The fold change is calculated as the ratio of the final value in the sham group to the value in each group (set as “1”).

### 4.4. Protein Preparation and Immunoblotting

Kidney samples were obtained for immunoblotting. Tissues were homogenized in RIPA buffer (Thermo Scientific, Waltham, MA, USA). Protein concentrations were determined using a protein assay kit (Bio-Rad, Hercules, CA, USA) with BSA as the standard. Fifty micrograms of total protein were subjected to 10–12% SDS–polyacrylamide gel electrophoresis. Proteins in the gel were then transferred to nitrocellulose membranes (Schleicher & Schuell, Dassel, Germany). Blots were probed with primary antibodies including polyclonal anti-transforming growth factor-β1 (TGF-β1) (diluted 1:500; sc146, Santa Cruz Biotechnology, Santa Cruz, CA, USA), anti-α-smooth muscle actin (α-SMA) (diluted 1:500; A5228, Sigma, St. Louis, MO, USA), NLRP-3 (Abcam, Cambridge, UK), c-Casp-1 (Abcam), and anti-β-actin (diluted 1:500; 3033, Cell Signaling Technology, Danvers, MA, USA) at 4 °C overnight. Blots were then incubated with a secondary antibody. Reactivity was visualized using an ECL kit. β-actin antibody (Sigma) served as a loading control. Densitometric data were used for quantitative analysis. The fold change is calculated as the ratio of the final value in the sham group to the value in each group (set as “1”).

### 4.5. Immunohistochemistry

An avidin-biotinylated-HRP (ABC) (Vector Laboratories, Burlingame, CA, USA) kit was used for immunohistochemistry studies, together with 5 μm thick paraformaldehyde-fixed, paraffin-embedded kidney sections. After incubation with 1% normal serum, sections were treated with primary antibodies of polyclonal Smad anti-F4/80 (diluted 1:100; 14-4801, ebioscience, San Diego, CA, USA) and monoclonal anti-α-SMA (diluted 1:500; A5228, Sigma, St. Louis, MO, USA) at 4 °C for 16 h. They were then washed with PBS (pH 7.4), incubated with a secondary antibody for 90 min, and then incubated with ABC for 60 min at room temperature. After rinsing sections with PBS, reactions were developed using 0.027% 3,3-diaminobenzidine tetrahydrochloride (Sigma, St. Louis, MO, USA) with 0.003% H_2_O_2_. These sections were then counterstained with hematoxylin to visualize cell nuclei. Next, sections were visualized by light microscopy. Digital images were captured and analyzed using NIS-Elements BR 3.2. Semiquantitative analysis was performed by counting the number of immunohistochemically stained positive cells per field in the renal tissue at ×400 magnification. The fold change is calculated as the ratio of the final value in the sham group to the value in each group (set as “1”).

### 4.6. Smad Quantitative Real-Time Polymerase Chain Reaction (PCR)

Kidney samples were obtained for quantitative real-time PCR. Total RNA was isolated from frozen kidney tissues using TRIzol (Invitrogen, Carlsbad, CA, USA). Purified RNAs were reverse transcribed into cDNAs using an iScript cDNA synthesis kit (Bio-Rad Laboratories, Hercules, CA, USA). Quantitative cDNA amplification was performed using a ViiA7 Real-Time System (Applied Biosystems Inc., Foster City, CA, USA), a Power SYBR Green PCR Master Mix (Applied Biosystems), and gene-specific primers for IL-1β, IL-18, TGF-1, and Snail. *GAPDH* was used as an internal control to normalize the quantity of RNA. The relative gene expression level in each sample was quantified using the 2^−ΔΔCt^ method. The following primer sequences were used: IL-1β, CTTCAGGCAGGCAGTATCACTCAT (F), TCTAATGGGAACGTCACACACCAG (R); IL-18, GCTGTGACCCTCTCTGTGAA (F), GGCAAGCAAGAAAGTGTCCT (R); TGF-1, TGCGCTTGCAGAGATTAAAA (F), CGTCAAAAGACAGCCACTCA (R); Snail1, CTTGTGTCTGCACGACCTGT (F), CTTCACATCCGAGTGGGTTT (R); and *GAPDH*, ACTCCACTCACGGCAAATTC (F), TCTCCATGGTGGTGAAGACA (R). The fold change is calculated as the ratio of the final value in the sham group to the value in each group (set as “1”).

### 4.7. Statistical Analysis

All statistical analyses were performed using GraphPad Prism software (version 9.0; GraphPad Software LLC, Boston, MA, USA). For more than two groups, mean values were compared using a one-way analysis of variance (ANOVA) with a comparison between groups by Tukey’s multiple comparison test (to compare all groups). A *p*-value < 0.05 was considered statistically significant.

## 5. Conclusions

We demonstrated that NLRP3 can promote TGF-β signaling, S-mad activation, and EMT in UUO kidneys and that PS can mitigate fibrosis by regulating these pathways. Specific therapies targeting CKD progression are very few in number. A better understanding of the mechanism involved in how PS can alleviate renal fibrosis might aid in the development of therapeutic interventions for patients with kidney diseases.

## Figures and Tables

**Figure 1 ijms-24-13171-f001:**
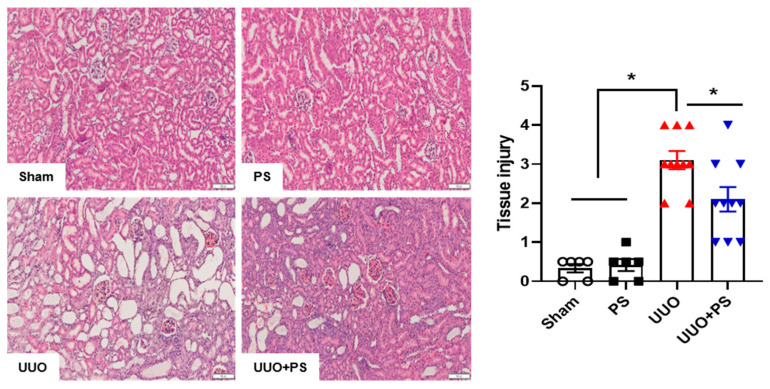
Effect of psoralen on UUO-induced tissue injury. Histological changes were examined using hematoxylin and eosin staining for sham control, psoralen alone, and UUO kidney with or without psoralen treatment (20 mg/kg). Tissue damage was quantified as described in the Materials and Methods section. Sham: no ureteral ligation; PS: only PS treatment; UUO: no PS treatment, but ureteral ligation; UUO + PS: PS treatment and ureteral ligation group. Data are presented as means ± SEM. * *p* < 0.05. Scale bar, 100 mm.

**Figure 2 ijms-24-13171-f002:**
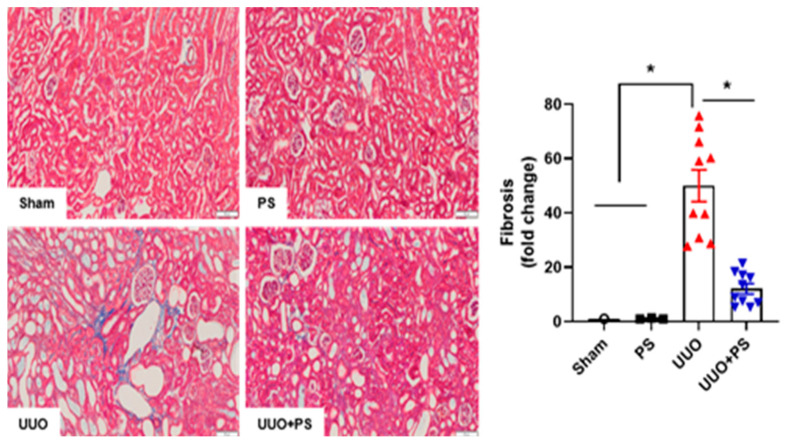
Effect of psoralen on UUO-induced renal fibrosis. Representative images of Masson’s trichrome staining for renal fibrosis. The severity of interstitial fibrosis was evaluated through MT staining and examined by densitometric quantification. Sham: no ureteral ligation; PS: only PS treatment; UUO: no PS treatment, but ureteral ligation; UUO + PS: PS treatment and ureteral ligation group. In this figure, each dot in the sham group represents the average value of three mice in that group as “1”. Data are presented as means ± SEM. * *p* < 0.05. Scale bar, 100 mm.

**Figure 3 ijms-24-13171-f003:**
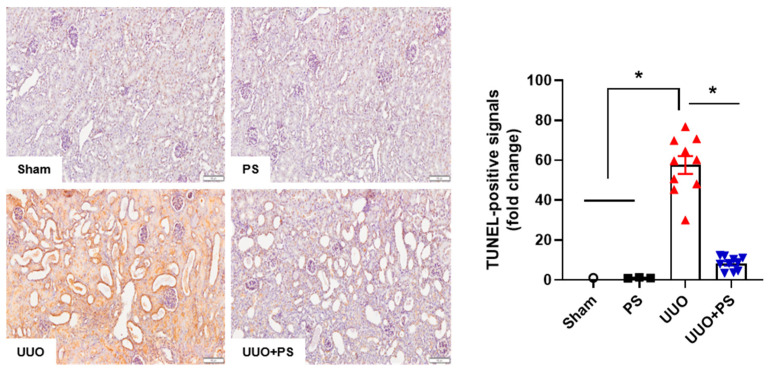
Effect of psoralen on UUO-induced apoptosis. Apoptotic cell death was examined by TUNEL assay. Quantitative analysis of TUNEL-positive cells was performed. Sham: no ureteral ligation; PS: only PS treatment; UUO: no PS treatment, but ureteral ligation; UUO + PS: PS treatment and ureteral ligation group. In this figure, each dot in the sham group represents the average value of three mice in that group as “1”. Data are presented as means ± SEM. * *p* < 0.05. Scale bar, 100 mm.

**Figure 4 ijms-24-13171-f004:**
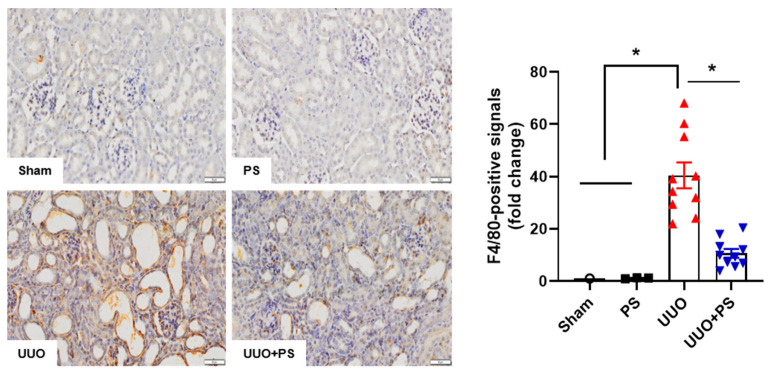
Psoralen ameliorates macrophage infiltration. F4/80 immunohistochemical staining was performed to verify macrophage infiltration. All signals were analyzed by densitometry. Sham: no ureteral ligation; PS: only PS treatment; UUO: no PS treatment, but ureteral ligation; UUO + PS: PS treatment and ureteral ligation group. In this figure, each dot in the sham group represents the average value of three mice in that group as “1”. Data are presented as means ± SEM. * *p* < 0.05. Scale bar, 50 μm.

**Figure 5 ijms-24-13171-f005:**
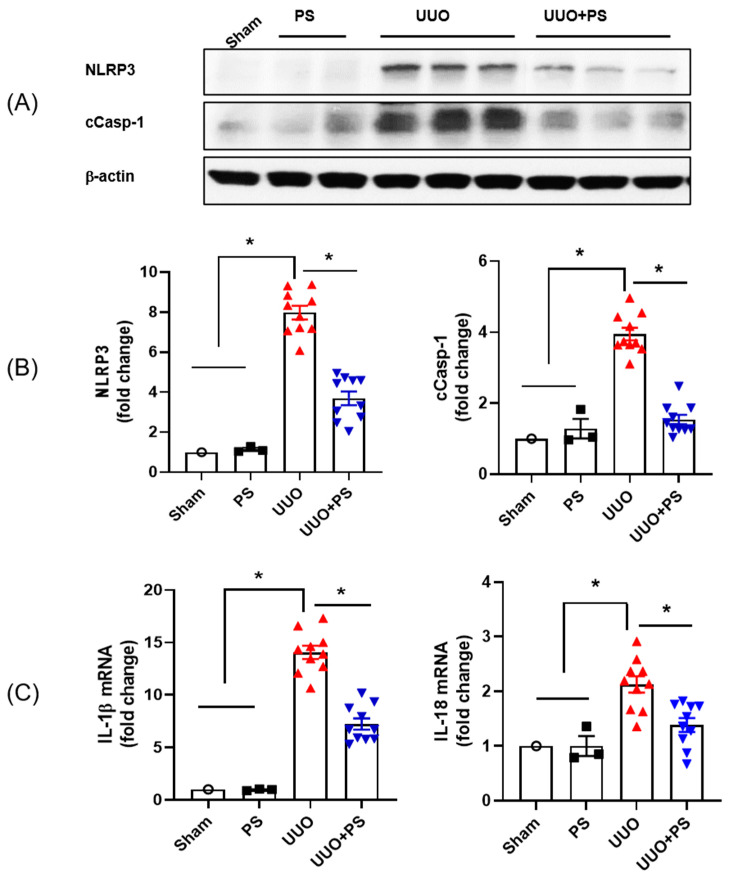
Effects of psoralen on UUO-induced renal inflammasome activation. Expression levels of NLRP-3 and cleaved caspase-1 (cCasp-1) were analyzed by Western blot. (**A**) Quantitative analyses of NLRP-3 and cCasp-1 were performed with results normalized to β-actin (**B**). mRNA expression levels of IL-1β and IL-18 were measured by quantitative real-time PCR (**C**). Double immunofluorescence staining for macrophage (green) and NLRP3 (red). White square boxes are representative double-positive signals for macrophage and NLRP3. Scale bar, 100 μm (**D**). In this figure, each dot in the sham group represents the average value of three mice in that group as “1”. Data are presented as mean ± SEM. * *p* < 0.05.

**Figure 6 ijms-24-13171-f006:**
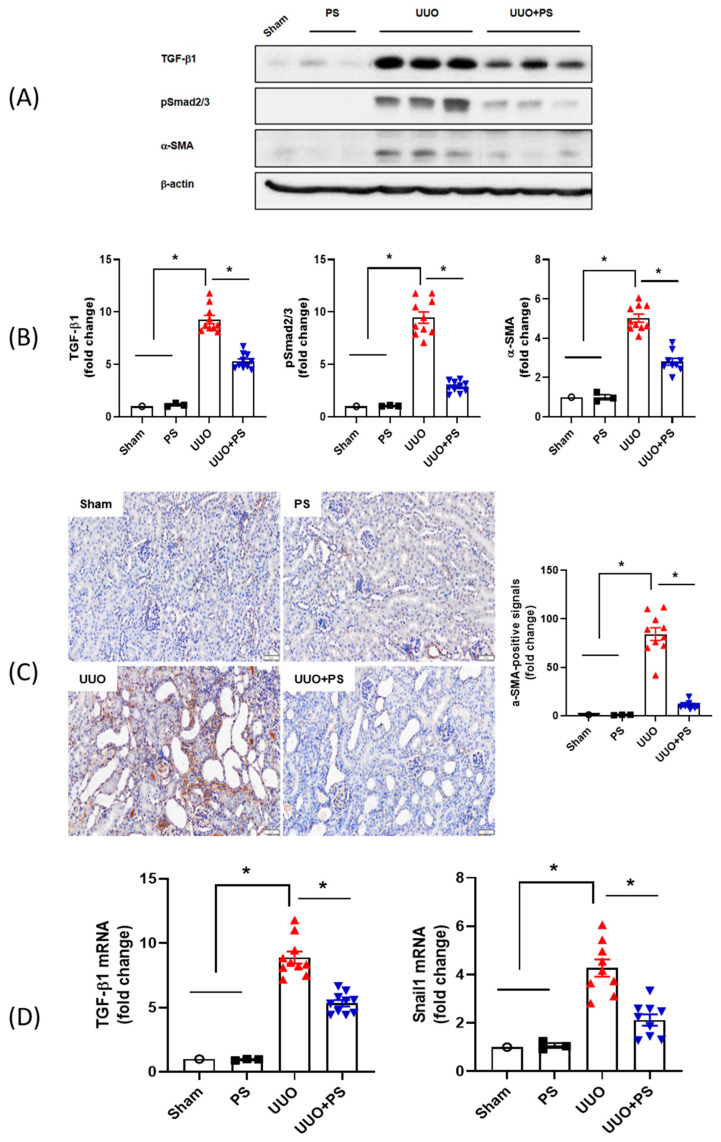
Effects of psoralen on UUO-induced renal fibrosis. Expression levels of TGF-β1, pSmad2/3, and α-SMA (**A**) were analyzed by immunoblotting. Quantitative analyses of TGF-β1, pSmad2/3, and α-SMA were performed. Results were normalized to levels of β-actin (**B**). α-SMA immunohistochemical staining was performed to verify fibrosis (**C**). mRNA expression levels of TGF-β1 and Snail were analyzed by quantitative real-time PCR (**D**). Scale bar, 50 μm. Data are presented as means ± SEMs. * *p* < 0.05. Sham: no ureteral ligation; PS: only PS treatment; UUO: no PS treatment, but ureteral ligation; UUO + PS: PS treatment and ureteral ligation group.

## Data Availability

Not applicable.

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
