# Peer review of "Psoralen Alleviates Renal Fibrosis by Attenuating Inflammasome-Dependent NLRP3 Activation and Epithelial–Mesenchymal Transition in a Mouse Unilateral Ureteral Obstruction Model"

_ijms, 2023, doi:10.3390/ijms241713171_

Round 1

Reviewer 1 Report

This study aimed to evaluate the effects of PS on the development and progression of renal fibrosis induced by UUO in a mouse model, suggesting that PS could be a promising therapeutic agent for managing chronic kidney disease (CKD). 

The authors have conducted this study with thorough preparation and execution; however, there is a need for them to address and clarify certain missing or incomplete aspects.

1. In abstract section, author described there are four groups: PS (n=5), PS+Sham (n=5), UUO (n=10), and UUO+PS (n=10). Then, section 4.1., there are four groups; a sham (n=5), PS (n=5), UUO (n=10), and UUP+PS (n=10). The author should make it clear. 

2. As they mentioned, there are 4 groups with n=5, 5, 10, 10, respectively. But there are Sham (n=6), PS (n=), UUO (n=10), UUO+PS (n=10) in Figure1. Then, there are Sham (n=1), PS (n=3) in Figure 2, 3, 4, and 5. In Fig 5, there are Sham group (n=3). Also, there are 9 data points of UUO or UUO+PS in Figure 5 and 6. It has to be very clear.

3. They used one-way ANOVA and Tukey's test for the statistical analysis. They may need to verify the statistical analysis due to control groups with n=1. 

4. Can UUO animal model represent the CKD? If yes, it would be beneficial for the author to explicitly address and describe this aspect in the paper.

5. This study exclusively utilized male mice. Considering the potential influence of sex hormones, is it possible that they might have affected the results of the study?

In result section, if it is the first time use, they need to use full name with abbreviation. MT staining stands for "Masson's trichrome".

Please double check sample numbers of each groups (group name as well).

Masson's, not Masson (pg 10)

400x magnification, not x400. (pg 10 and 11)

remove the space in unit from 5 µ m (pg 10)

Reviewer 2 Report

Dear authors,

you present data supporting that psoralen mitigates NLRP3 inflammasome downstream signaling.

My major comments are:

How specific is psoralen-mediated NLRP3 activation? Do you have relevant data demonstrating that while NLRP3 is inactivated, psoralen fails to induce the pathways under study?

Interstitial nephritis is a non-specific reaction in post-renal acute kidney injury.  Furthermore, tissue injury results in non-specific NLRP3 activation. These are associative events and do not prove causality. Do you have the capacity to prove that macrophage infiltration occurs dependent on NLRP3 activation?  Furthermore, how does psoralen induce NLRP3 activation? Could you for example provide data concerning psoralen action(s) from NLRP3 knockouts or Il1b knockouts? If mice are not available, do you have the capacity to perform experiments in cell culture systems presenting an abolished NLRP3 activity?

Minor comments:

Somehow all images seem distended and of poor quality. Please modify.

I would like to see the whole Western blots films and not only the truncated version.

For the comments above, I am of the opinion that your manuscript requires a major rewrite.

All of my best regards.

The English language used in this manuscript is appropriate.

Round 2

Reviewer 1 Report

Authors addressed well about all concerns.